# Citrate Dialysate with and without Magnesium Supplementation in Hemodiafiltration: A Comparative Study Versus Acetate

**DOI:** 10.3390/ijms25158491

**Published:** 2024-08-03

**Authors:** Diana Rodríguez-Espinosa, Elena Cuadrado-Payán, Naira Rico, Mercè Torra, Rosa María Fernández, Gregori Casals, María Rodríguez-García, Francisco Maduell, José Jesús Broseta

**Affiliations:** 1Nephrology and Renal Transplantation, Hospital Clínic of Barcelona, 08036 Barcelona, Spain; dmrodriguez@clinic.cat (D.R.-E.); ecuadrado@clinic.cat (E.C.-P.); fmaduell@clinic.cat (F.M.); 2Biochemistry and Molecular Genetics Department, Biomedical Diagnostic Center, Hospital Clínic of Barcelona, 08036 Barcelona, Spain; nrico@clinic.cat (N.R.); mtorra@clinic.cat (M.T.); rmfernandez@clinic.cat (R.M.F.); casals@clinic.cat (G.C.); mrodriguezg@clinic.cat (M.R.-G.)

**Keywords:** citrate, acetate, dialysate, magnesium, hemodiafiltration, hemodialysis, divalent metals, chelation

## Abstract

The choice of dialysate buffer in hemodialysis is crucial, with acetate being widely used despite complications. Citrate has emerged as an alternative because of its favorable effects, yet concerns persist about its impact on calcium and magnesium levels. This study investigates the influence of citrate dialysates (CDs) with and without additional magnesium supplementation on CKD-MBD biomarkers and assesses their ability to chelate divalent metals compared to acetate dialysates (ADs). A prospective crossover study was conducted in a single center, involving patients on thrice-weekly online hemodiafiltration (HDF). The following four dialysates were compared: two acetate-based and two citrate-based. Calcium, magnesium, iPTH, iron, selenium, cadmium, copper, zinc, BUN, albumin, creatinine, bicarbonate, and pH were monitored before and after each dialysis session. Seventy-two HDF sessions were performed on eighteen patients. The CDs showed stability in iPTH levels and reduced post-dialysis total calcium, with no significant increase in adverse events. Magnesium supplementation with CDs prevented hypomagnesemia. However, no significant differences among dialysates were observed in the chelation of other divalent metals. CDs, particularly with higher magnesium concentrations, offer promising benefits, including prevention of hypomagnesemia and stabilization of CKD-MBD parameters, suggesting citrate as a viable alternative to acetate. Further studies are warranted to elucidate long-term outcomes and optimize dialysate formulations. Until then, given our results, we recommend that when a CD is used, it should be used with a 0.75 mmol/L Mg concentration rather than a 0.5 mmol/L one.

## 1. Introduction

The 5-year mortality rate for patients undergoing hemodialysis (HD) is 50%, with most deaths attributed to cardiovascular causes, especially non-atherothrombotic events [1]. Hemodiafiltration (HDF) offers a survival advantage over HD [2,3], but a significant residual risk remains. Inflammation, endothelial damage, vascular calcification, and myocardial structural changes are significant factors contributing to this high mortality rate. Therefore, it is essential for nephrologists to seek strategies consistently to improve the cardiovascular outlook for these patients.

The dialysate plays a crucial role in HD treatment. Its components determine the direction and final concentration of solute diffusive exchange through the semipermeable membrane. One crucial component is the dialysate buffer. Although acetate is the most commonly used buffer, its use can elevate blood levels beyond the normal physiological range, leading to reduced dialysis tolerance [4,5], inflammation [6,7], oxidative stress [8], hemodynamic instability [4,9], and cardiovascular alterations [10,11]. As a result, researchers have studied alternative buffers to replace acetate, with citrate being the most extensively researched and widely used.

Citrate offers several advantages beyond being an acetate-free dialysate. It has been associated with lower levels of inflammatory markers [6,12,13,14], improved dialysis tolerance [4,15,16,17], less dialyzer clotting [18], and potential reduction of vascular calcification [10,19]. Despite these benefits, its use has not yet become widespread in centers, primarily because of concerns and uncertainties about the clinical effects of the citrate chelation of calcium and magnesium. For instance, 1 mmol/L of citrate can bind 1.5 mmol/L of both cations [20]. It is through its calcium-chelating capacity that this dialysate manages to reduce heparin expenditure and dialyzer coagulation, as calcium plays a role in the coagulation pathway [21].

Several observational studies have found a correlation between magnesium levels and mortality in patients undergoing HD. Concentrations of magnesium ranging from 2.5 to 2.8 mg/dL have been linked to improved survival [22,23]. The use of citrate instead of acetate can cause a marked reduction in blood magnesium levels [24]. Therefore, magnesium supplementation during citrate dialysis could help prevent hypomagnesemia, potentially providing a survival advantage for these patients. However, this hypothesis is yet to be confirmed by interventional studies [25].

Citrate is also believed to reduce inflammation by binding iron or copper, reducing their ability to act as cofactors in activating the complement pathway [17,26]. However, these beneficial effects have failed to prove better survival outcomes [24,27]. Nevertheless, no data are available on this subject, and it is unknown whether the supplementation of another divalent metal will affect the amount of citrate-bonded calcium levels or other metals.

Some manufacturers offer a higher concentration of calcium and magnesium in citrate dialysates (CDs) with the intention to supplement them and, thus, counteract citrate’s ability to bind these two metals. Acetate dialysates (ADs) are usually available with a calcium concentration of either 1.5 (AD-Ca1.5) or 1.25 (AD-Ca1.25) mmol/L and a fixed one with a magnesium concentration of 0.5 mmol/L. In contrast, CDs are available with a calcium supplement of 1.5, 1.65, or 1.75 mmol/L and a variable magnesium supplement of either 0.5 (CD-Mg0.5) or 0.75 mmol/L (CD-Mg0.75).

This study aims to examine and compare the impact of CDs with or without magnesium supplementation versus ADs in HDF on chronic kidney disease–mineral bone disorder (CKD-MBD) biomarkers and other divalent metals with a potential role in inflammation.

## 2. Results

Seventy-two HDF sessions were performed on 18 patients, with each patient receiving one session with each dialysate. The dialysates’ compositions can be found in Table 1. Thirteen patients were male (72%), the median age was 80 (69–83) years, 12 had an AVF as vascular access (66.7%), and the others used a tunneled catheter. The causes of CKD were urologic (6, 33.3%), diabetes mellitus (3, 16.7%), glomerulonephritis (3, 16.7%), autosomal dominant polycystic kidney disease (1, 5.6%), multiple myeloma (1, 5.6%), cardiorenal (1, 5.6%), and unknown (3, 16.7%). Low-molecular-weight heparin was the most used anticoagulant (55.5%), followed by unfractionated heparin (27.8%), and no anticoagulant was used in 16% of the sessions.

There were no differences in the blood flow (Qb), dialysate fluid flow (Qd), dialysis session length, ultrafiltration, recirculation, replacement and total treated blood volume, the anticoagulation dose used, lowest relative blood volume, or pre- and post-dialysis systolic and diastolic blood pressures. There were also no significant differences between pre- and post-dialysis creatinine, blood urea nitrogen (BUN), or albumin, nor in their reduction ratios. Also, there were no statistically significant differences in Kt or blood pressure, nor were there changes in pre- and post-dialysis pH, bicarbonate, or total carbon dioxide with either dialysate. Refer to Table 2 for further details.

### 2.1. Calcium and CKD-MBD Biomarkers

There were no differences in pre-dialysis values of total and ionized calcium, intact parathyroid hormone (iPTH), or phosphorus. Regarding post-dialysis values, we found that total calcium increased with AD-Ca1.5, CD-Mg0.5 (1.16 ± 0.66 and 0.28 ± 0.51 mg/dL, respectively, both with a *p*-value < 0.001), and CD-Mg0.75 (0.29 ± 0.55 mg/dL, *p* = 0.006), while it remained similar with AD-Ca1.25 (−0.07 ± 0.48 mg/dL, *p* = 0.224). Though there was an increase in total calcium levels with both CD formulations after dialysis, these increases were significantly lower compared with AD-Ca1.5 (*p* < 0.001 for both). In contrast, post-dialysis ionized calcium significantly decreased with both CD-Mg0.5 (−0.11 ± 0.1 mmol/L, *p* < 0.001) and CD-Mg0.75 (−0.07 ± 0.09 mmol/L, *p* = 0.006), increased with AD-Ca1.5 (0.13 ± 0.08 mmol/L, *p* < 0.001), and remained stable with AD-Ca1.25 (0.02 ± 0.08 mmol/L, *p* = 0.224). See Figure 1.

Regarding iPTH levels, we found that they remained stable, without significant differences between the pre- and post-dialysis values with CD-Mg0.5 (228.78 ± 27.14 and 214.28 ± 34.97, *p* = 0.33), CD-Mg0.75 (229.83 ± 27.21 and 269.5 ± 43.53, *p* = 0.28), or AD-Ca1.25 dialysates (244.83 ± 31.83 and 207.78 ± 44.79, *p* = 0.33). It was only significantly lowered with AD-Ca1.5 (238.44 ± 26.46 and 74.28 ± 11.93, *p* < 0.001). Notably, there were four instances with post-dialysis iPTH values above 600 pg/mL, where three were from the same patient. There was a significant correlation between the pre- and post-dialysis iPTH delta and pre- and post-dialysis ionized and total calcium delta values (Figure 2) (R^2^ = 0.378, *p* < 0.001 and R^2^ = 0.452, *p* < 0.001, respectively). There was no correlation between the delta iPTH values and the delta magnesium (R^2^ = 1.38 × 10^−5^, *p* = 0.975).

Finally, post-dialysis phosphorus levels significantly decreased compared with pre-dialysis ones (−2.28 ± 2.1, −2.47 ± 0.18, −2.53 ± 0.18, and −2.19 ± 0.37, with all comparisons with a *p*-value < 0.001) without differences among the four studied dialysates (AD-Ca1.5, AD-Ca1-25, CD-Mg0.5, and CD-Mg0.75, respectively). The main results of the CKD-MBD variables are summarized in Table 3.

### 2.2. Magnesium

There were no significant differences between magnesium pre-dialysis values. However, its post-dialysis levels were significantly reduced with AD-Ca1.5 (−0.22 ± 0.07, *p* = 0.006), AD-Ca1.25 (−0.21 ± 0.07, *p* = 0.006), and CD-Mg0.5 (−0.39 ± 0.07, *p* < 0.001). The latter was significantly lower than the former (*p* < 0.001). In the case of the magnesium-supplemented citrate dialysate (CD-Mg0.75), post-dialysis magnesium levels increased significantly from the pre-dialysis ones (0.23 ± 0.07, *p* = 0.005). These results are detailed in Table 3.

### 2.3. Cadmium, Selenium, Copper, Zinc, and Iron

There were no differences among the pre-dialysis values for any of these metals. When analyzing by each dialysate (AD-Ca1.5, AD-Ca1.25, CD-Mg0.5, and CD-Mg0.75), selenium (−11.33 ± 3.47, *p* < 0.001; −9.19 ± 1.99, *p* = 0.005; −6.14 ± 1.65, *p* = 0.002; −11.21 ± 1.71; *p* < 0.001), copper (−8.56 ± 1.84, −7.72 ± 1.66, −8.64 ± 1.88, −12.01 ± 1.93; *p* < 0.001 for all of them), and iron (−14.07 ± 19.09, *p* = 0.01; −14.24 ± 22.02, *p* = 0.06; −16.63 ± 15.69, *p* < 0.001; −11.46 ± 13.93, *p* < 0.001) serum levels significantly decreased post-dialysis with all dialysates with no differences among them. Cadmium and zinc remained similar after dialysis, with all four dialysates having no differences among them. More detailed information is presented in Figure 1.

## 3. Discussion

Many experts have expressed concerns about citrate’s ability to bind calcium and cause hypocalcemia [28]. However, several studies, including our own, have shown that citrate does not significantly increase the risk of symptomatic hypocalcemia [15]. In our research, the only dialysate that reduced post-dialysis total calcium concentration was AD-Ca1.25, while the others showed no differences.

We found that ionized calcium levels were higher after dialysis when using ADs and slightly but significantly lower when using CDs. Using the latter may help decrease the amount of calcium gained during dialysis treatments. Despite the decrease in ionized calcium, we did not observe a significant increase in adverse events reported with CDs compared to ADs. We did find a correlation between iPTH and ionized calcium plasma levels, as previously published [29], with no specific difference when comparing dialysates with each other. In fact, reductions in ionized calcium are minimal or non-existent, as shown in previous publications [10,30], and ionic calcium levels actually increase about 30 to 60 min after the session ends, as calcium citrate is metabolized. This release of calcium occurs in hepatocytes, making hepatic failure an important factor in citrate accumulation and toxicity [20,31]. Interestingly, the use of CD has been linked with a lower risk of vascular calcification because of the lack of calcium loading [19,32,33].

This study found that post-dialysis blood levels of iPTH did not significantly differ from pre-dialysis levels, except when using acetate as the dialysate buffer with a calcium concentration of 1.5 mmol/L, as noted by Argilés et al. [30]. We found that the differences in iPTH levels between acetate and CD are due to the suppression of iPTH caused by hypercalcemia when using an AD, rather than an iPTH stimulation caused by citrate-induced reductions in calcium levels. These results suggest that the difference observed in post-dialysis iPTH between the AD and CD is due to an inhibitory effect driven by hypercalcemia when patients used the acetate buffer containing 1.5 mmol/L of calcium, rather than iPTH stimulation by citrate or relative hypocalcemia, as reported in previous works [34,35].

The significance of studying different magnesium levels in the dialysate has evolved. A lower magnesium dialysis concentration was believed to have biochemical advantages by correcting hypermagnesemia and maintaining average erythrocyte potassium concentrations [36,37]. However, later research found that higher serum magnesium levels may be linked to a lower mortality rate [22,23], making the previous arguments less critical.

Hypomagnesemia has been associated with a higher risk of death in hemodialysis patients [25]. Though not exactly clear, the observed increase in mortality could be attributed to short-term effects such as a cardiac arrhythmia accompanied by rapid variations in potassium blood concentration, changes in pH, and QT prolongation [24] or more long-term ones like hypomagnesemia accelerating vascular calcification, especially in CKD patients [38]. Some researchers have suggested specific target magnesium values of 2.5 and 2.8 mg/dL, which have been found to improve survival rates [22,23]. However, this finding has only been observed in HD patients, not in HDF, especially when acetate is used as the primary dialysis buffer. Only one study by Pérez-García et al. [24] included some HDF patients to compare survival rates between a CD and AD. Their study identified a higher occurrence of hypomagnesemia in patients using the CD with the standard magnesium concentration of 0.5 mmol/L compared to those using acetate. The study suggested that hypomagnesemia might have been a confounding factor in their mortality outcomes, masking improved survival rates of the CD versus AD. It is possible that the survival effect of calcification reduction induced by citrate was neutralized by hypomagnesemia. Our study found that using a magnesium dialysate concentration of 0.5 mmol/L, regardless of whether acetate or citrate were used as buffers, resulted in post-dialysis magnesium levels <2 mg/dL. This was overcome by supplementing with 0.75 mmol/L magnesium concentration, which increased post-dialysis levels compared with pre-dialysis ones. This is an important finding as it achieves levels above the cut-off value for the survival advantage established by Pérez-García et al. of 2.1 mg/dL [24].

The ability of citrate to bind to other divalent ions has led to the hypothesis that certain metallic elements, such as iron or copper, could explain the beneficial effect of CDs on inflammation or endothelial damage [17,27]. This study could not demonstrate this hypothesis, as the levels of these elements did not differ among dialysates.

Additionally, evidence suggests the stability of selenium and zinc in HDF patients [39] and a link between cadmium levels and inflammation and malnutrition in dialysis patients [40]. However, the chelating effect of citrate on these elements in the hemodialysis setting has not been fully described [41,42,43,44]. In our results, only selenium, copper, and iron were significantly removed by dialysis. On the other hand, copper and zinc levels in the blood remained unchanged compared to pre-dialysis levels. Further research is needed in this area.

This study has some limitations. Firstly, we cannot draw definite conclusions about the long-term effects of different calcium concentrations on CKD-MBD biomarkers since we only evaluated these dialysates in a few sessions. Additionally, using an AD with 1.25 mmol/L of calcium is known to increase PTH levels and worsen secondary hyperparathyroidism over time. In our study, we only collected blood samples 10 min after dialysis and did not assess the complete kinetics of calcium and magnesium; therefore, we cannot conclude whether these findings were sustained during the interdialysis period. However, it is known that citrate is metabolized entirely in bicarbonate from 30 to 60 min after the end of the dialysis session, releasing calcium and magnesium ions. Although it is unlikely, further assessment at the 3- and 6-month marks would be necessary to confirm the consistency in this effect on calcium chelation with an AD with 1.25 mmol/L of calcium. Thirdly, this study involved a small group of patients from one center, and we only measured the total concentration of some of the divalent metals, not the ionized forms. Therefore, there may be a concentration difference that our assay was unable to detect. Finally, in our cohort, all patients were on post-dilution HDF. It is generally understood that for a given concentration gradient between blood and dialysate, the calcium balance in pre-dilution HDF will be lower than in high-flux HD, and often negative, especially with higher ultrafiltration rates [45]. Therefore, when switching from HD or post-dilution HDF to pre-dilution HDF, an increase in dialysate calcium of approximately 0.5 mEq/L is recommended [46]. Further studies will need to assess the calcium balance in those situations. Despite these limitations, this study provides valuable insights into the behavior and effects of CDs, which can help tailor hemodialysis treatments to better suit our patients.

## 4. Materials and Methods

### 4.1. Setting

This is a prospective crossover study carried out in a single center. All patients from our center’s hemodialysis morning shift were considered for inclusion. Included patients had to be on a thrice-weekly online HDF program for at least three months, had less than 250 mL of urine output, had an available well-functioning vascular access (capable of >350 mL/min of prescribed blood flow), and had provided informed consent. Patients with a scheduled living donor kidney transplant within the next month, severe (<7.0 mg/dL) or symptomatic hypocalcemia, taking oral magnesium supplements, or with active cancer or infection were excluded. This study was approved by the local Ethics Committee and was conducted according to the Declaration of Helsinki.

### 4.2. Materials

In this study, four types of dialysates were compared. Two of them were acetate-based, including SmartBag^®^ 211.5 (AD-Ca1.5) and SmartBag^®^ 211.25 (AD-Ca1.25), with a calcium concentration of 1.5 and 1.25 mmol/L, respectively. The other two were citrate-based, including SmartBag^®^ CA 211.5 (CD-Mg0.5) and SmartBag^®^ CA 211.5-0.75 (CD-Mg0.75), both with 1.5 mmol/L calcium concentrations but differing in 0.5 and 0.75 mmol/L magnesium concentrations, respectively. Fresenius Medical Care, Bad-Homburg, Germany, manufactures all these dialysates. Details on each dialysate’s composition can be found in Table 1. Each patient was randomized to one of four groups, where each session was conducted with one dialysate on the mid-dialysis weekday in the following order:(a)Week 1: SmartBag 211.5, week 2: SmartBag 211.25, week 3: SmartBag CA 211.5, week 4: SmartBag CA 211.5-0.75(b)Week 1: SmartBag 211.25, week 2: SmartBag CA 211.5, week 3: SmartBag CA 211.5-0.75, week 4: SmartBag 211.5.(c)Week 1: SmartBag CA 211.5, week 2: SmartBag CA 211.5-0.75, week 3: SmartBag 211.5, week 4: SmartBag 211.25.(d)Week 1: SmartBag CA 211.5-0.75, week 2: SmartBag 211.5, week 3: SmartBag 211.25, week 4: SmartBag CA 211.5.

Every studied subject utilized Fresenius 6008 CAREsystem™ dialysis monitors and FX CorDiax™ 60 dialyzers (Fresenius Medical Care, Bad Homburg v.d.H., Germany).

### 4.3. Variables

Demographic and medical history were obtained from electronic health records. During each dialysis session, the following variables were recorded: dialyzer used, anticoagulant type and dose, Qb, Qd, real dialysis time, Kt, total treated volume, replacement volume, ultrafiltration, vascular access recirculation, minimal relative blood volume (obtained from the Fresenius Medical Care^®^ Blood Volume Monitor^®^, Fresenius Medical Care, Bad Homburg v.d.H., Germany), and blood pressure before and after each HDF session. Also, patients were encouraged to report the appearance of any symptoms (e.g., muscle cramps, headaches, or tingling) during the hemodialysis session and were asked on the next session if they had presented any unusual symptoms at home.

Blood samples were taken directly from the vascular access before the start and 10 min after finishing each dialysis. The following values were analyzed: blood urea nitrogen, creatinine, albumin, ionized and total calcium, phosphorus, iPTH measured by the Immulite Decentralised Procedure assay, magnesium, pH, total carbon dioxide, and bicarbonate, as well as iron, selenium, zinc, copper, cadmium pre- and post-dialysis values, along with their reduction ratios. Post-dialysis concentrations were adjusted for ultrafiltration [47].

### 4.4. Statistical Analysis

A repeated measures ANOVA with a Greenhouse–Geisser correction was used to determine differences in the mean concentrations at different time points and among the dialysates. Post hoc analyses with a Bonferroni adjustment were then used to determine statistically significant differences between pairs. The results are expressed as the arithmetic mean ± standard deviation, and *p* < 0.05 was considered statistically significant. Analyses were performed using SPSS software version 23 (SPSS, Chicago, IL, USA).

## 5. Conclusions

There have been concerns about citrate’s ability to bind calcium and magnesium, but our study and others suggest that CDs could help reduce calcium buildup during dialysis and have a positive impact on vascular calcification. We discovered that the differences in iPTH levels between acetate and CDs are due to the suppression of iPTH caused by hypercalcemia when using ADs, rather than an iPTH stimulation caused by citrate-induced reductions in calcium levels. Additionally, the connection between magnesium levels and mortality in hemodialysis patients emphasizes the importance of adjusting magnesium concentrations in dialysates when using citrate. By correcting magnesium levels, we may now be able to observe increased survival in this population, but more studies are needed for this purpose. Current nephrology practices need to focus on personalized treatments to enhance cardiovascular outcomes in dialysis patients, and citrate seems to show promise in reducing the high residual cardiovascular risk. Until then, given our results, we recommend that when a CD is used, it should be used with a 0.75 mmol/L Mg concentration rather than a 0.5 mmol/L one.

## Figures and Tables

**Figure 1 ijms-25-08491-f001:**
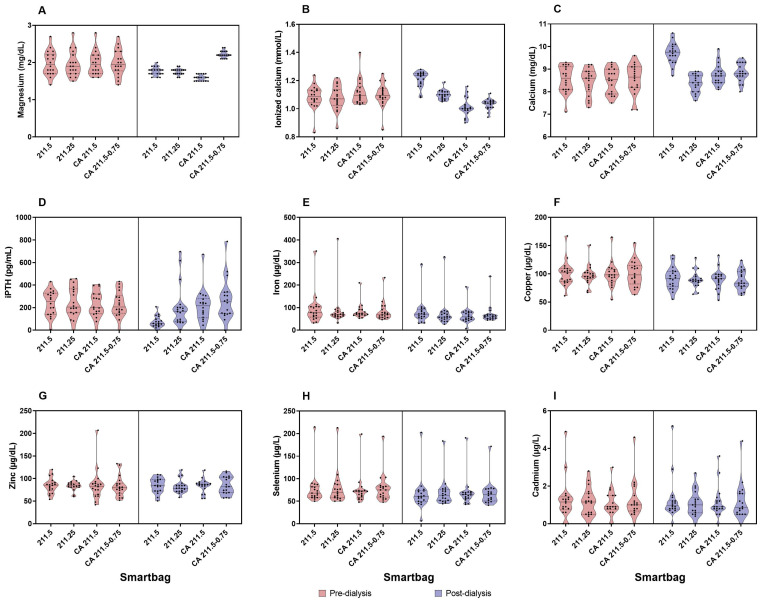
Violin plots of pre- and post-dialysis concentrations with each dialysate for (**A**). magnesium, (**B**). ionized calcium, (**C**). total calcium, (**D**). PTH, (**E**). iron, (**F**). copper, (**G**). zinc, (**H**). selenium, and (**I**). cadmium. iPTH: intact parathyroid hormone. SmartBag^®^ 211.25: acetate dialysate with a calcium concentration of 1.25 mmol/L and magnesium concentration of 0.5 mmol/L, SmartBag^®^ 211.5: acetate dialysate with a calcium concentration of 1.5 mmol/L and magnesium concentration of 0.5 mmol/L, SmartBag^®^ CA 211.5: citrate dialysate with a calcium concentration of 1.5 mmol/L and magnesium concentration of 0.5 mmol/L, and SmartBag^®^ CA 211.5-0.75: citrate dialysate with a calcium concentration of 1.5 mmol/L and magnesium concentration of 0.75 mmol/L.

**Figure 2 ijms-25-08491-f002:**
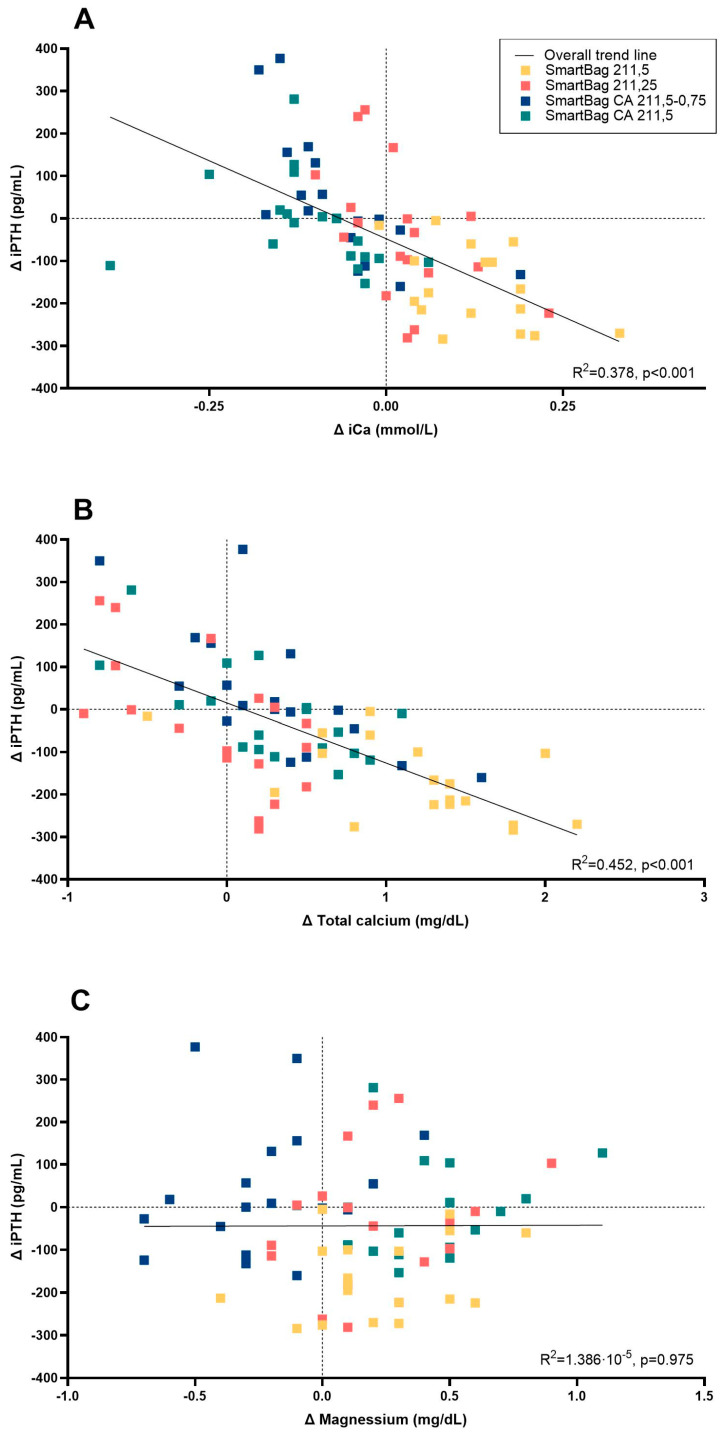
Correlation of the delta between the pre- and post-dialysis intact PTH blood concentration with iCa (**A**), total calcium (**B**), and magnesium (**C**) blood values. iCa, ionized calcium; iPTH, intact parathyroid hormone.

**Table 1 ijms-25-08491-t001:** Dialysates’ compositions.

Components	SmartBag 211.25	SmartBag 211.5	SmartBag CA 211.5	SmartBag CA 211.5-0.75
Sodium (mmol/L)	138	138	138	138
Potassium (mmol/L)	2	2	2	2
Calcium (mmol/mL)	1.25	1.5	1.5	1.5
Magnesium (mmol/mL)	0.5	0.5	0.5	0.75
Chloride (mmol/mL)	108.5	109	109	109.5
Acetate (mmol/L)	3	3	-	-
Citrate (mmol/L)	-	-	1	1
Glucose (g/L)	1	1	1	1
Bicarbonate (mmol/L)	32	32	32	32
Osmolarity (mosm/L)	290.8	291.55	290	290

**Table 2 ijms-25-08491-t002:** Dialysis sessions’ adequacies and safety parameters.

Variable	SmartBag 211.25	SmartBag 211.5	SmartBag CA 211.5	SmartBag CA 211.5-0.75	*p* Value
Blood flow (mL/min, mean ± SD)	449 ± 1	444 ± 4	447 ± 3	446 ± 3	0.364
Dialysis session (min, mean ± SD)	284.33 ± 4.31	284.5 ± 4.17	285.44 ± 4.37	285.33 ± 4.08	0.152
Total ultrafiltration (L, mean ± SD)	1.79 ± 0.82	1.98 ± 0.7	1.97 ± 1.05	2.03 ± 0.79	0.618
Treated blood volume (L, mean ± SD	125.13 ± 8	126.36 ± 8.16	126.26 ± 8.51	125.82 ± 9.47	0.426
Kt, (n ± SD)	70.07 ± 6.09	70.97 ± 7.19	71.64 ± 7.47	71.48 ± 6.46	0.348
BUN RR (%, mean ± SD)	84.62 ± 3.08	85.23 ± 4	85.6 ± 3.04	84.79 ± 2.94	0.345
Creatinine RR (%, mean ± SD)	78.05 ± 4.44	77.84 ± 4.68	78.89 ± 4.23	77.9 ± 4.7	0.276
Albumin RR (%, mean ± SD)	8.53 ± 5.84	9.54 ± 5.88	9.06 ± 6.27	10.67 ± 3.88	0.228
ΔpH (mean ± SD)	0.09 ± 0.01	0.07 ± 0.02	0.07 ± 0.01	0.06 ± 0.01	0.394
ΔHCO_3_ (mean ± SD)	4.89 ± 0.42	4.09 ± 0.65	4.72 ± 0.45	4.58 ± 0.38	0.573
ΔTCO_2_ (mean ± SD)	4.61 ± 0.47	3.78 ± 0.51	3.22 ± 0.45	3.39 ± 0.36	0.051
Replacement volume (L, mean ± SD)	31.7 ± 4.92	31.22 ± 5.8	32.05 ± 5.78	31.35 ± 5.21	0.604
Recirculation (%, mean ± SD)	17.22 ± 5.7	16.56 ± 4.74	15.28 ± 4.73	15.56 ± 2.85	0.187
BVM nadir (kg/m^2^ mean ± SD)	90.75 ± 3.47	90.55 ± 2.86	90.46 ± 4.07	90.54 ± 3.51	0.988
Pre-dialysis SBP (mmHg, mean ± SD)	123.7 ± 25.6	126.8 ± 27.9	120.9 ± 23.5	125.9 ± 25.6	0.433
Post-dialysis SBP (mmHg, mean ± SD)	112.1 ± 21.2	115.8 ± 16.2	106.5 ± 15	105.2 ± 13.6	0.054
Pre-dialysis DBP (mmHg, mean ± SD)	60 ± 13.2	60 ± 11.3	58.3 ± 13.1	61.2 ± 11.5	0.698
Post-dialysis DBP (mmHg, mean ± SD)	54.7 ± 10.4	56.1 ± 7	53.8 ± 10.7	53.1 ± 6.8	0.54

BUN: blood urea nitrogen, BVM: blood volume monitoring, DBP: diastolic blood pressure, HCO_3_: bicarbonate, RR: reduction ratio, SBP: systolic blood pressure, TCO_2_: total carbon dioxide.

**Table 3 ijms-25-08491-t003:** CKD-MBD parameters.

Variable	SmartBag 211.25	SmartBag 211.5	SmartBag CA 211.5	SmartBag CA 211.5-0.75	*p* Value
Total calcium (mg/dL)					
Pre-dialysis	8.44 ± 0.59	8.54 ± 0.59	8.49 ± 0.56	8.55 ± 0.67	0.139
Post-dialysis	8.38 ± 0.4	9.71 ± 0.47	8.77 ± 0.46	8.84 ± 0.41	<0.001
Variation	−0.07 ± 0.48	1.16 ± 0.66	0.28 ± 0.51	0.29 ± 0.55	<0.001
Ionized calcium (mmol/L)					
Pre-dialysis	1.08 ± 0.09	1.08 ± 0.09	1.12 ± 0.09	1.1 ± 0.09	0.114
Post-dialysis	1.1 ± 0.03	1.21 ± 0.06	1.01 ± 0.07	1.03 ± 0.04	<0.001
Variation	0.02 ± 0.08	0.13 ± 0.08	−0.11 ± 0.1	−0.07 ± 0.09	<0.001
Parathormone (pg/mL)					
Pre-dialysis	244.83 ± 31.83	238.44 ± 26.46	228.78 ± 27.14	229.83 ± 27.21	0.785
Post-dialysis	207.78 ± 44.79	74.28 ± 11.93	214.28 ± 34.97	269.5 ± 43.53	<0.001
Variation	−37.06 ± 36.96	−164.17 ± 21.73	−12.5 ± 26.02	39.67 ± 35.55	<0.001
Phosphorus (mg/dL)					
Pre-dialysis	3.86 ± 0.21	3.81 ± 0.23	3.82 ± 0.21	3.85 ± 0.18	0.984
Post-dialysis	1.39 ± 0.06	1.53 ± 0.07	1.28 ± 0.06	1.66 ± 0.33	0.424
Variation	−2.47 ± 0.18	−2.28 ± 2.1	−2.53 ± 0.18	−2.19 ± 0.37	0.622
Magnesium (mg/dL)					
Pre-dialysis	1.97 ± 0.08	1.98 ± 0.8	1.98 ± 0.08	1.99 ± 0.08	0.862
Post-dialysis	1.76 ± 0.02	1.77 ± 0.03	1.59 ± 0.02	2.22 ± 0.02	<0.001
Variation	−0.21 ± 0.07	−0.22 ± 0.07	−0.39 ± 0.07	0.23 ± 0.07	<0.001

## Data Availability

The data supporting the findings of this study are available on GitHub (https://github.com/Broseta/Citrate-dialysate.git, accessed on 15 July 2024).

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
