# Peer review of "Citrate Dialysate with and without Magnesium Supplementation in Hemodiafiltration: A Comparative Study Versus Acetate"

_ijms, 2024, doi:10.3390/ijms25158491_

Round 1

Reviewer 1 Report

Comments and Suggestions for Authors

In this paper the authors explore the role of dialysate citrate for post-dialysis calcium and magnesium levels and whether a higher dialysate magnesium concentration affects post-dialysis magnesium levels.

I find the paper clinically relevant and well-written with interesting data. I have a few comments and suggestions.

1) Methods:

Which dialysis machine was used for study ?

Are the pre- and post-dialysis blood samples drawn from a separate venous line or from the AV fistula ?

You mention measurements of pH, bicarbonate and tCO2 (section 4.4), but are these data included ?

2) Results:

Figure 1: It should be more clear whether it is blood or plasma concentrations of metals (zinc, iron, cupper etc.). These figures are however very small and the text on the axes difficult to read.

Consider the use of decimals. Use max 1 decimal for all BP data. No decimals for blood flows etc. 

It is not clear why you present both total and ionized calcium ?   I suppose ionized is the important parameter and what controls PTH (as also show in Figure 2).  what should the reader learn from using both parameters ?

Table 2: line 1: What are these numbers. They seem to represent the blood flow (not total calcium) ?   

Figure 2: There are no units on the axes. I suggest you mark the symbols according to the dialysate composition (preferentially using 4 different colours). Also, there is no information whether the slope is the same for all 4 dialysate compositions.

Did you detect any influence of delta-magnesium on changes in PTH ?

3) Discussion:

As mentioned above I suggest to analyse and comment on the delta-iCa - PTH relation for each dialysate composition.

You state that a dialysate magnesium above 2 mg/dl could be important for patient outcome. This may be achieved  using a dialysate magnesium of 0.75 M, but does this result in a sustained higher level or is it short-lasting ? 

I am not sure how the statement on line 277-279 is clear from the data ?  a reduction in iCa using citrate dialysate will also increase post-dialysis PTH ?

Comments on the Quality of English Language

Overall I find the language good; only minor improvements can be made.

Author Response

In this paper the authors explore the role of dialysate citrate for post-dialysis calcium and magnesium levels and whether a higher dialysate magnesium concentration affects post-dialysis magnesium levels.

I find the paper clinically relevant and well-written with interesting data. I have a few comments and suggestions.

1) Methods:

Comment 1: Which dialysis machine was used for study?

Response 1: Thank you for your comments. We have added the brand and model of the dialysis machine used in the Methods section: “This study utilized Fresenius® 6008 CAREsystem® dialysis monitors and FX CorDiax® 60 Fresenius dialyzers.”

Comment 2: Are the pre- and post-dialysis blood samples drawn from a separate venous line or from the AV fistula ?

Response 2: Thank you for your comments. Samples for this study were obtained directly from the vascular access used for dialysis rather than from peripheral blood. This aspect has been clarified in the methods section: “Blood samples were taken directly from the vascular access before the start and 10 minutes after finishing each dialysis”.

Your comment made us realize that we had not properly included the information about the patients' vascular accesses. We have now corrected this in the Results section.

Comment 3: You mention measurements of pH, bicarbonate and tCO2 (section 4.4), but are these data included?

Response 3: Thank you for noticing this. As we found no differences, we did not consider to put the data in the manuscript. However, taken into consideration that negative results must also be published, this information has been added to the results section and to Table 1: “Also, there were no statistically significant differences in Kt, blood pressure, nor changes in pre- and post-dialysis pH, bicarbonate or total carbon dioxide with either dialysate. Refer to Table 1 for further details.”

2) Results:

Comment 4: Figure 1: It should be more clear whether it is blood or plasma concentrations of metals (zinc, iron, cupper etc.). These figures are however very small and the text on the axes difficult to read.

Response 4: We appreciate your feedback. We have specified that metals were measured in plasma, and we have modified the font size in the figure to make it more legible as well as changed the grey scale for colors to improve visibility.

Comment 5: Consider the use of decimals. Use max 1 decimal for all BP data. No decimals for blood flows etc.

Response 5: We have modified the values on Table 1 as you suggested.

Comment 6: It is not clear why you present both total and ionized calcium?   I suppose ionized is the important parameter and what controls PTH (as also show in Figure 2).  what should the reader learn from using both parameters ?

Response 6: We agree with the reviewer on the greater importance of ionized calcium (iCa) in the modulation of parathyroid hormone (PTH). However, it is also important to consider total calcium (tCa) measurements. Total calcium includes both the biologically active ionized calcium and the complexed calcium (CiCa). This bound calcium is released approximately 20-30 minutes post-dialysis, converting back to the ionized form as represented in the following reaction:

Ca3(C6H5O7)2  +  9 O2  6 CO2  +  6 HCO3-  +  3 Ca2+  +  2 H2O

This delayed release can impact calcium homeostasis and subsequently PTH levels. Thus, including both parameters allows us to provide a comprehensive picture of calcium dynamics before, during, and after dialysis.

Comment 7: Table 2: line 1: What are these numbers. They seem to represent the blood flow (not total calcium) ?  

Response 7: You are correct. This is a typo. We used Table 1 as a model for Table 2 and mistakenly left the blood flow values. That line was meant to remain empty. We have fixed this.

Comment 8: Figure 2: There are no units on the axes. I suggest you mark the symbols according to the dialysate composition (preferentially using 4 different colours). Also, there is no information whether the slope is the same for all 4 dialysate compositions

Response 8: We have specified the measurement units in the figure and figure legend and assigned different colors to each dialysate. We retained the overall trend line because drawing the four individual trend lines would introduce noise and reduce clarity, without providing additional valuable information, as the trends are quite similar.

Comment 9: Did you detect any influence of delta-magnesium on changes in PTH?

Response 9: We did not find any correlation between these two variables. We have added this to the results section: “There was no correlation between the delta iPTH values with the delta total calcium and magnesium values.”

3) Discussion:

Comment 10: As mentioned above I suggest to analyse and comment on the delta-iCa - PTH relation for each dialysate composition.

Response 10: Thank you for your insightful suggestion. We did not observe specific differences between the dialysates, as shown in Table 2. We have addressed this finding in the Discussion section: “We did find a correlation between iPTH and ionized calcium plasma levels as previously published [30], with no specific difference when comparing dialysates between each other.”

Comment 11: You state that a dialysate magnesium above 2 mg/dl could be important for patient outcome. This may be achieved using a dialysate magnesium of 0.75 M, but does this result in a sustained higher level or is it short-lasting ?

Response 11: Thank you for your question. We are currently unable to determine whether the increase in magnesium levels is sustained or short-lasting. We have added this limitation to the manuscript: “In our study, we only collected blood samples 10 minutes after dialysis and did not assess the complete kinetics of calcium and magnesium, therefore we cannot conclude whether these findings were sustained during the interdialysis period.”

Comment 12: I am not sure how the statement on line 277-279 is clear from the data ?  a reduction in iCa using citrate dialysate will also increase post-dialysis PTH ?

Response 12: Thank for your comment. We agree that the sentence was not clear enough. We have modified the phrase into: “We found that the differences in iPTH levels between acetate and citrate dialysates are due to the suppression of iPTH caused by hypercalcemia when using acetate dialysate, rather than an iPTH stimulation caused by citrate-induced reductions in calcium levels.”

Reviewer 2 Report

Comments and Suggestions for Authors

Rodríguez-Espinosa et al provided interesting data on optimization of dialysate formulations. The study is clinically relevant. The methodology and results are clearly presented. I have minor comments regarding the Discussion part that need to be addressed.

Here are some academic questions for discussion based on the provided text:

1. Please highlight the part of discussion explaining how do different magnesium concentrations in dialysate influence the biochemical outcomes in hemodialysis (HD) patients.

2. What are the potential mechanisms by which higher serum magnesium levels might reduce mortality in HD patients?

3. How do citrate and acetate as dialysis buffers affect magnesium levels and overall patient outcomes and what are the potential pathways linking hypomagnesemia to increased mortality in HD patients?

 4. What are the clinical implications of your finding that a magnesium dialysate concentration of 0.75 mmol/L achieves higher post-dialysis magnesium levels compared to pre-dialysis levels?

5. How might hypomagnesemia act as a confounding factor in studies comparing survival rates between citrate and acetate dialysates?

 6. What further research is needed to elucidated the relationship between magnesium concentration in dialysate and patient survival?

7. What protocols could be developed to monitor and adjust magnesium levels in HD patients to optimize their outcomes?

Author Response

Rodríguez-Espinosa et al provided interesting data on optimization of dialysate formulations. The study is clinically relevant. The methodology and results are clearly presented. I have minor comments regarding the Discussion part that need to be addressed.

Here are some academic questions for discussion based on the provided text:

Comment 1: Please highlight the part of discussion explaining how do different magnesium concentrations in dialysate influence the biochemical outcomes in hemodialysis (HD) patients.

Response 1: Thank you for your time and comments. We have addressed the issue of magnesium concentration in the dialysate in the 4th and 5th paragraph of the Discussion section:

“The significance of studying different magnesium levels in the dialysate has evolved. A lower magnesium dialysis concentration was believed to have biochemical advantages by correcting hypermagnesemia and maintaining average erythrocyte potassium concentrations [37,38]. However, later research found that higher serum magnesium levels may be linked to a lower mortality rate [22,23], making the previous arguments less critical.

Hypomagnesemia has been associated with a higher risk of death in hemodialysis patients [25]. Though not exactly clear, the observed increase in mortality could be at-tributed to a cardiac arrhythmia accompanied by rapid variations in potassium blood concentration, changes in pH, and by QT prolongation [24]. Some researchers have suggested specific target magnesium values of 2.5 and 2.8 mg/dL, which has been found to improve survival rates [22,23].”

Comment 2: What are the potential mechanisms by which higher serum magnesium levels might reduce mortality in HD patients?

Response 2: Thank you for your question. We have added the mechanism in the discussion section: “Though not exactly clear, the observed increase in mortality could be attributed to a cardiac arrhythmia accompanied by rapid variations in potassium blood concentration, changes in pH, and by QT prolongation [24].”

Comment 3: How do citrate and acetate as dialysis buffers affect magnesium levels and overall patient outcomes and what are the potential pathways linking hypomagnesemia to increased mortality in HD patients?

Response 3: In the introduction and in the discussion, we mention that citrate, unlike acetate, has the capacity to bind to divalent metals, as calcium and magnesium. In regard to the mortality observed HD patients with hypomagnesemia, we have added the line: “Though not exactly clear, the observed increase in mortality could be attributed to a cardiac arrhythmia accompanied by rapid variations in potassium blood concentration, changes in pH, and by QT prolongation [24].” In the discussion section.

Comment 4: What are the clinical implications of your finding that a magnesium dialysate concentration of 0.75 mmol/L achieves higher post-dialysis magnesium levels compared to pre-dialysis levels?

Response 4: We appreciate your feedback. We have modified the Conclusion section to make this clearer:

“There have been concerns about citrate's ability to bind calcium and magnesium, but our study and others suggest that citrate dialysates could help reduce calcium buildup during dialysis and have a positive impact on vascular calcification. We discovered that the differences in iPTH levels between acetate and citrate dialysates are due to the suppression of iPTH caused by hypercalcemia when using acetate dialysate, rather than an iPTH stimulation caused by citrate-induced reductions in calcium levels. Additionally, the connection between magnesium levels and mortality in hemodialysis patients emphasizes the importance of adjusting magnesium concentrations in dialysates when using citrate. It's possible that the survival effect of calcification reduction induced by citrate was neutralized by hypomagnesemia. By correcting magnesium levels, we may now be able to observe increased survival in this population, but more studies are needed for this purpose. Current nephrology practices need to focus on personalized treatments to enhance cardiovascular outcomes in dialysis patients, and citrate seems to show promise in reducing the high residual cardiovascular risk.”

Comment 5: How might hypomagnesemia act as a confounding factor in studies comparing survival rates between citrate and acetate dialysates?

Response 5: Thank you for making us remark this in the article. It may act as a confounding factor because hypomagnesemia is associated with increased mortality, therefore by correcting it, we may now properly see if citrate truly renders a survival benefit or not. As stated in the previous answer, we believe that the modifications made to the Conclusion section have improved clarity on this subject.

Comment 6: What further research is needed to elucidated the relationship between magnesium concentration in dialysate and patient survival?

Response 6: In the conclusion we have tackled this question. In the final lines we state that: “…By correcting magnesium levels, we may now be able to observe increased survival in this population, but more studies are needed for this purpose. Current nephrology practices need to focus on personalized treatments to enhance cardiovascular outcomes in dialysis patients, and citrate seems to show promise in reducing the high residual cardiovascular risk.”

Comment 7: What protocols could be developed to monitor and adjust magnesium levels in HD patients to optimize their outcomes?

Response 7: This is a very interesting question. Our study group believes in a personalized dialysis, with values adjusted to each patient’s requirements. We have published papers on the personalized adjustment of bicarbonate, sodium, and, we will certainly add magnesium to that list, once we gain more information and experience on its use with citrate. Until then, given our results, we recommend that, when a citrate dialysate is used, it should be used with a 0.75 mmol/L Mg concentration rather than a 0.5mmol/L one. We have added this last sentence in the Conclusion section and in the abstract.